# Rare Filaggrin Variants Are Associated with Pustular Skin Diseases in Asians

**DOI:** 10.3390/ijms25126466

**Published:** 2024-06-12

**Authors:** Luca Lo Piccolo, Wasinee Wongkummool, Phatcharida Jantaree, Teerada Daroontum, Suteeraporn Chaowattanapanit, Charoen Choonhakarn, Warayuwadee Amornpinyo, Romanee Chaiwarith, Salin Kiratikanon, Rujira Rujiwetpongstorn, Napatra Tovanabutra, Siri Chiewchanvit, Chumpol Ngamphiw, Worrachet Intachai, Piranit Kantaputra, Mati Chuamanochan

**Affiliations:** 1Centre of Multidisciplinary Technology for Advanced Medicine (CMUTEAM), Faculty of Medicine, Chiang Mai University, Chiang Mai 50200, Thailand; lopiccolo.l@cmu.ac.th (L.L.P.); wasinee.won@cmu.ac.th (W.W.); phatcharida.j@cmu.ac.th (P.J.); 2Department of Pathology, Faculty of Medicine, Chiang Mai University, Chiang Mai 50200, Thailand; mewteerada@gmail.com; 3Division of Dermatology, Department of Medicine, Faculty of Medicine, Khon Kaen University, Khon Kaen 40002, Thailand; csuteeraporn@yahoo.com (S.C.); c_choonhakarn@yahoo.com (C.C.); 4Division of Dermatology, Department of Internal Medicine, Khon Kaen Hospital, Ministry of Public Health, Khon Kaen 40002, Thailand; warayuwadee.a@gmail.com; 5Division of Infectious Diseases and Tropical Medicine, Department of Internal Medicine, Faculty of Medicine, Chiang Mai University, Chiang Mai 50200, Thailand; rchaiwar@gmail.com; 6Division of Dermatology, Department of Internal Medicine, Faculty of Medicine, Chiang Mai University, Chiang Mai 50200, Thailand; s.kiratikanon@gmail.com (S.K.); rujira.r.330@gmail.com (R.R.); ntovanabutra@gmail.com (N.T.); drsiri2010@gmail.com (S.C.); 7National Biobank of Thailand, National Center for Genetic Engineering and Biotechnology (BIOTEC), Pathum Thani 12120, Thailand; chumpol.nga@gmail.com; 8Center of Excellence in Medical Genetics Research, Faculty of Dentistry, Chiang Mai University, Chiang Mai 50200, Thailand; worrachet.intachai@gmail.com (W.I.); dentaland17@gmail.com (P.K.); 9Division of Pediatric Dentistry, Department of Orthodontics and Pediatric Dentistry, Faculty of Dentistry, Chiang Mai University, Chiang Mai 50200, Thailand

**Keywords:** reactive pustular eruptions, AOID, pustular psoriasis, whole-exome sequencing, filaggrin

## Abstract

Reactive pustular eruptions (RPEs) can manifest in a variety of conditions, including pustular psoriasis (PP) and adult-onset immunodeficiency syndrome due to anti-interferon-γ autoantibody (AOID). These RPEs can be attributed to different causes, one of which is genetic factors. However, the genetic basis for pustular skin diseases remains poorly understood. In our study, we conducted whole-exome sequencing on a cohort of 17 AOID patients with pustular reactions (AOID-PR) and 24 PP patients. We found that 76% and 58% of the AOID-PR and PP patients, respectively, carried rare genetic variations within the filaggrin (FLG) gene family. A total of 12 out of 21 SNPs on FLG had previously received clinical classifications, with only p.Ser2706Ter classified as pathogenic. In contrast, none of the FLG3 SNPs identified in this study had prior clinical classifications. Overall, these variations had not been previously documented in cases of pustular disorders, and two of them were entirely novel discoveries. Immunohistochemical analysis of skin biopsies revealed that FLG variants like p.Ser860Trp, p.Gly3903Ter, p.Gly2440Glu, and p.Glu2133Asp caused reductions in FLG levels similar to the pathogenic FLG p.Ser2706Ter. These results highlight rare FLG variants as potential novel genetic risk factors contributing to pustule formation in both AOID and PP.

## 1. Introduction

Adult-onset immunodeficiency syndrome due to anti-interferon-γ autoantibody (AOID) and pustular psoriasis (PP) are distinct conditions with different underlying causes and clinical presentations, yet they share some commonalities. For instance, both AOID and PP can present with reactive pustular eruptions (RPEs), which are characterized by the sudden onset of sterile pustules on the skin [1,2,3]. The pustules in both AOID and PP are caused by an abnormal accumulation of leukocytes, mainly neutrophils, and cellular debris [4,5]. Both AOID and PP are characterized by immune system dysregulation, resulting in the formation of pustular eruptions. In AOID, this dysregulation is primarily attributed to the presence of anti-interferon-γ autoantibodies, whereas in PP, it is linked to the dysregulation of the IL-36 signaling pathway [6,7].

Recent research indicates that genetic factors may also contribute to the pathogenesis of both AOID and PP. Notably, these conditions share common genetic risk factors, such as mutations in *SERPINA1*, *SERPINB3*, and *TGFBR2*, suggesting a potential role for serine/threonine kinase variants in the underlying mechanisms of both disorders [2,4,8]. Additional genetic risk factors have been associated with PP, including mutations in genes involved in the IL-36 signaling pathway, such as *IL36RN*, *CARD14*, *AP1S3*, and *MPO*. Conversely, AOID has shown associations with *HLA-DRB1* and *HLA-DQB1* alleles [1,3,9].

Overall, the genetics of pustular skin reactions seem to be multifactorial, and both genetic and environmental factors likely play roles in disease development and progression. Despite the recent progression in this field, the genetic architecture of pustular skin reactions is still not fully elucidated.

The rarity of severe skin diseases, their diverse cutaneous and extracutaneous manifestations, and the similarity of their symptoms to other dermatological conditions present significant challenges in promptly diagnosing and treating affected individuals, especially in the case of PP. Current laboratory tests employed for PP diagnosis typically are not specific to PP and primarily focus on assessing inflammatory markers, along with clinical and histopathologic features of PP. Establishing distinct subtypes of PP and AOID through genetic analyses can offer valuable guidance for therapeutic decisions, enabling patients to achieve more precise and timely treatment outcomes.

Our study sought to reveal common genetic elements shared by RPEs in AOID and PP, with the intention of elucidating the fundamental mechanisms underpinning these disorders and pinpointing potential targets for future precision medicine interventions.

## 2. Results and Discussion

### 2.1. Demographic Data

A total of 41 blood samples were collected in this study, including 17 samples from AOID patients with pustular reactions (AOID-PR) and 24 samples from PP patients (Figure 1). The ages of onset of the pustular reactions in AOID-PR and PP were 52.9 ± 12 and 40.4 ± 24 years, respectively. Females accounted for 8/17 (47.1%) in the AOID-PR group and 18/24 (75%) in the PP group. Among the PP patients, 18 (75%) had generalized PP, 4 (16.7%) had acrodermatitis continua of Hallopeau, and 2 (8.3%) had palmoplantar psoriasis.

### 2.2. Several Rare Unreported Genetic Variants of Filaggrin Are Associated with RPEs

Next, we examined the whole-exome sequencing (WES) data of patients with AOID-PR and PP to investigate the genetic factors associated with RPEs. This process involved selecting variants from each participant based on specific criteria in comparison to public population databases, including 1000 G, gnomAD exomes (gnomADe), gnomAD genomes (gnomADg) v.2.1.1, and the Thai Reference Exome variants database (T-REx) (AF 1 kg < 0.01; gnomADe < 0.05; gnomADg < 0.05; and genotype quality > 20). Taking into account population specificity, to further filter out variants, these pre-selected variants were then compared to an additional in-house database consisting of 315 unrelated WES samples from a cohort of patients with dental anomalies, none of whom presented underlying skin diseases. Specifically, we focused on rare variants (AF < 0.01) as potential candidates, and we ranked them according to their prevalence among the patient groups. To enhance our selection process, we leveraged additional information provided by VarSome (varsome.com) to prioritize genes with molecular functions that were pertinent to the phenotype under investigation.

Our analysis revealed that 76% of the AOID-PR patients (13 out of 17) carried genetic variations in two genes from the filaggrin family, *FLG* and *FLG3* (also known as *HRNR*). Interestingly, a similar pattern emerged from the whole-exome study of the PP cohort, where 58% of the patients (14 out of 24) had genetic variations in *FLG* or *FLG3* (Table 1). None of the patients with FLG or FLG3 variants had an atopic background. Patients lacking variants in filaggrins displayed a high degree of genetic diversity. This observation supports the suggested hypothesis that the genetics of RPEs are indeed multifactorial. As this study aimed to identify common genetic elements, our focus shifted towards filaggrin.

The spectrum of the identified FLG and FLG3 variants also exhibited significant variability, with the p.Arg843Ter FLG variant (rs141263661) being the only one shared among two AOID-PR cases (Figure 2A,B). Overall, 47% of the AOID-PR patients and 33% of the PP patients were carriers of FLG variants, while 35% of the AOID-PR patients and 29% of the PP patients were carriers of FLG3 variants. Small percentages, 0.6% of AOID-PR patients and 0.4% of PP patients, were carriers of both FLG and FLG3 variants (Table 1).

A few variants were identified in previous ClinVar records and were subsequently assessed for their putative clinical significance using VarSome. For FLG, three were classified as benign (B), eight as likely benign (LB), and one as pathogenic (P), while no data were available for eight variants. Notably, for eight FLG variants and all identified FLG3 variants there was a lack of available data that would facilitate a putative clinical classification (refer to Appendix A).

Notably, two variants, namely p.Gly3903Ter in FLG and p.Ser2830PhefsTer3 in FLG3, had not been previously reported in the SNP database (Figure 2, red marks). Additionally, in silico analysis predicted that both p.Gly3903Ter and p.Ser2830PhefsTer3 would have substantial impacts on the functions of the FLG and FLG3 proteins, as summarized in Appendix A.

The proteins encoded by FLG genes are involved in aggregating keratin intermediate filaments in the mammalian epidermis [10,11]. FLG proteins are initially synthesized as a polyprotein precursor, profilaggrin, which is localized in keratohyalin granules and subsequently proteolytically processed into individual functional filaggrin molecules [12,13]. Despite the critical role played by filaggrin molecules in forming a physical barrier and supporting the skin, several functional genetic variations impacting FLG have been uncovered.

Strikingly, both common and population-specific loss-of-function (LoF) variants have been documented, with reports emerging from European [14] and Asian populations [15,16]. To date, only a limited number of LoF variants have been observed in individuals of African descent, specifically among African Americans [17]. These FLG LoF variants result in impaired skin-barrier function, which, in turn, increases the risk of various complex skin disorders. This impairment leads to symptoms such as dry, itchy, and inflamed skin, along with elevated transepidermal water loss, collectively representing significant risk factors for numerous complex skin disorders [18,19].

Notably, prior in-depth examination of the *FLG* locus at the haplotype level uncovered distinctive signs of a recent selective sweep in Asia. This event resulted in increased frequency of a specific haplotype group known as the “Huxian haplogroup” within Asian populations [20]. Within this Huxian haplogroup, numerous functional variants of the *FLG* and *FLG3* genes have been pinpointed. However, despite the presence of some common genetic *FLG* variants with significant phenotypic effects in the Asian population, an exploration of a whole-exome sequencing (WES) database containing 1092 Thai individuals [21] revealed that the 18 *FLG* and 15 *FLG3* single-nucleotide polymorphisms (SNPs) presented here had exceedingly low allele frequencies (AF < 0.003) (Table 1).

Taken together, these findings demonstrate that rare genetic variations in the FLG gene family are linked to RPEs in PP and AOID and could have potential as diagnostic tools.

### 2.3. The Newly Identified Variants Have an Impact on FLG Expression

Considering the crucial role of filaggrin in skin health and its previous link to reduced FLG function in various inflammatory skin disorders, we sought to assess the influence of FLG variants on pustule eruptions. As part of this investigation, we conducted immunohistochemistry on paraffin-embedded skin biopsies to quantify the FLG levels in both AOID-PR and PP skin samples. The immunoreactivity of FLG in five AOID-PR and two PP patients who carried FLG variants was compared to that of healthy donors of Thai nationality who did not present AD or other skin conditions (Figure 3 and Figure 4 and Appendix A). Among the patients examined, it is noteworthy that three out of five AOID-PR patients and both PP patients displayed reduced FLG immunoreactivity. Intriguingly, the novel FLG variant p.Gly3903Ter, identified in this study, and the previously unclassified p.Glu2133Asp and p.Gly2440Glu variants resulted in declines in FLG expression similar to the pathogenic p.Ser2706Ter FLG carrier. Additionally, FLG p.Ser860Trp, which had previously been classified as benign, also showed reduced FLG immunoreactivity (Figure 3 and Figure 4).

To determine if the reductions in FLG levels were solely attributable to the pustular eruptions, we conducted immunohistochemistry on skin biopsy samples from AOID-PR and PP patients who did not possess any FLG variants (FLG-WT). In these cases, the FLG immunoreactivity in the skin biopsies closely resembled that of healthy individuals (Figure 3 and Figure 4).

These findings strongly suggest that p.Gly3903Ter, p.Glu2133Asp, p.Gly2440Glu, and p.Ser2706Ter may impact FLG and the skin barrier and could indicate potential contributions to the pustular eruptions observed in both AOID and PP. However, we observed significant variability in severity assessments among evaluators, which rendered it impossible to establish any potential correlation between the FLG variants and the severity of pustulosis. Consequently, we highly recommend that future studies aiming to investigate such correlations adopt validated clinical standards, such as those proposed in [22].

We acknowledge the limited size of the control cohort of Thai nationality presented in this study, which may have affected the robustness of the findings. Future studies should aim to include a broader and more diverse set of control samples to improve the generalizability and accuracy of the results. In addition, AGEP should ideally be included in this analysis. However, skin biopsies and diagnostic confirmation tests are not routinely performed to confirm AGEP in our clinical practice.

Finally, these data expand our understanding of the role of FLG in skin diseases and contribute to the growing body of knowledge regarding the genetic factors underlying pustule formation in both AOID and PP. Furthermore, in terms of therapeutic implications, utilizing treatment modalities such as l-histidine and vitamin D receptor agonists to enhance FLG may help reduce the severity of pustulosis in a specific population subgroup, as observed in AD [23,24].

## 3. Patients and Methods

### 3.1. Study Subjects

A multicenter retrospective study of all patients with AOID at Maharaj Nakorn Chiang Mai Hospital, Khon Kaen University’s Srinagarind Hospital, and Khonkaen Hospital in Thailand was conducted from January 2005 to June 2020. Diagnoses of AOID needed to fulfill all of the following criteria: (i) a history of opportunistic infections (OIs) that represented a cell-mediated immunity (CMI) defect; (ii) exclusion of other immunosuppressed states, such as HIV, malignancy, or receiving immunosuppressive drugs; and (iii) demonstration of antibodies against IFN-γ using an enzyme-linked immunosorbent assay (ELISA).

The pustular reactions in the AOID patients were confirmed based on the clinical manifestation of multiple tiny non-follicular pustules with the exclusion of infection by any of the following microbiologic methods: (i) staining; (ii) culture; and (iii) molecular techniques. All participants denied family and personal histories of psoriasis. No culprit drugs responsible for acute generalized exanthematous pustulosis (AGEP) at the onset of pustular eruption were identified.

Pustular psoriasis (PP) was clinically diagnosed by a board-certified dermatologist in accordance with the guidelines proposed by the European Rare and Severe Psoriasis Expert Network (ERASPEN) [25]. It was categorized into the generalized form (i.e., generalized PP) and the localized form, which included palmoplantar pustulosis (PPP) and acrodermatitis continua of Hallopeau (ACH).

This study was approved by the institutional review boards of the Faculty of Medicine, Chiang Mai University (study code: MED-2563-07128); Srinagarind Hospital, Khon Kaen University (study code: HE631340); and Khonkaen Hospital, Khon Kaen (study code: KEF65019).

### 3.2. Molecular Analysis and Assessment of Rare Variants

Genomic DNA was extracted from 17 AOID patients and 24 GPP patients to be exome-sequenced with a SureSelect V6 UTR (Agilent Technologies, USA) captured library kit from Macrogen, Seoul, Korea. Following WES variant analysis (Appendix A), post-processing analysis was performed on the resulting FASTQ data based on best practices using GATK 3.8 [26] to call germline short variants based on human reference sequence builds GRCh37 or hg19 in the genomic variant call format (gVCF). The gVCF files were called genotypes using CombineGVCFs and GenotypeGVCFs. Then, we evaluated the variant quality using GATK VQSR and filtered the genotyped variants with stringent filtering criteria, including a genotype quality (GQ) greater than 20 and a read depth (DP) greater than 10, using VCFTools [27]. The short variants (SNVs and INDELs) that passed these filtering criteria included 8,548,085 loci, which were annotated by VEP build110 [28] with the dbNSFP v4.4a plugin [29] integrated with custom public population databases, including 1000 G, gnomAD exomes (gnomADe), gnomAD genomes (gnomADg) v.2.1.1) [30], and the Thai Reference Exome variants database (T-REx) [21]. Only variants that were predicted to have high and moderate impacts by VEP, with rare allele frequencies compared to 1000 G, gnomADe, and gnomADg (less than 0.01) and T-REx (less than 0.05), were considered as candidate pathogenic variants for each sample. Furthermore, those filtered variants were then compared with an in-house database of 315 unrelated whole-exome sequencing (WES) samples from diverse rare dental diseases of patients with no pre-existing dermatological conditions. The candidate genes were ranked by the number of samples that contained candidate pathogenic variants. Two possible candidate genes associated with pustule skin diseases were identified.

### 3.3. Immunohistochemistry

The skin tissue samples were fixed in a 10% neutral-buffered formalin solution and embedded in paraffin. For histopathological analysis, formalin-fixed paraffin-embedded (FFPE) tissues were cut at a thickness of 5 µm and stained with hematoxylin and eosin (H&E) utilizing conventional laboratory techniques. To assess the staining for filaggrin (FLG), the FFPE blocks were cut into 5 μm sections and subjected to a 1 h heating process at 60 °C in a dry oven. To remove the paraffin, xylene was applied to the sections, followed by gradual rehydration with graded ethanol in water. Antigen retrieval was performed on a Benchmark ULTRA automated slide stainer for 32 min at 37 °C using CC1 (prediluted, pH 8.0) antigen retrieval solution (Ventana Medical Systems, Roche Group, Tucson, AZ, USA). IHC staining was performed using a Ventana BenchMark ULTRA autostainer(Ventana Medical Systems, Roche Group, Tucson, AZ, USA) using a standard protocol. The sections were then incubated with the primary antibody (Mouse Monoclonal anti-filaggrin antibody, clone FLG01, 1:200 dilution, Cat# MA5-13440, Thermofisher, Waltham, MA, USA). An Ultraview Universal DAB IHC detection kit was used for the visualization reaction (12 min). All sections were then counterstained with a hematoxylin nuclear stain and a blue reagent. The slides were covered with coverslips using a mounting material after being cleaned and dehydrated in graded ethanol and xylene. We used healthy human skin as a positive control for the filaggrin (FLG) antibody. The control group exclusively consisted of individuals without atopic dermatitis (AD).

For image analysis, an Aperio Scanscope CS2 whole-slide scanner (Leica Biosystems, Nussloch, Germany), together with Aperio ImageScope version 12.4.6.5003 software (Leica Biosystems, Wetzlar, Germany), was used to capture digital pictures of the IHC-stained slides. We selected the entire epidermal region, including the pustule, using the pen tool in the Aperio ImageScope software (version 12) and then measured the IHC staining using the positive pixel count algorithm (version 9) embedded in the Aperio ImageScope software. The output of the positive pixel count was defined as follows: In = total intensity of negative signal, Iwp = total intensity of weak positive signal, Ip = total intensity of positive signal, Isp = total intensity of strong positive signal, and NTotal = total number (positive + negative). An intensity score was calculated to determine the overall signal intensity for all individual tissue spots of IHC antibody staining using the sum of the intensity values for all negative, weak positive, positive, and strong positive pixels (In + Iwp + Ip + Isp) divided by the total number of pixels (Ntotal).

### 3.4. Data Analysis

Statistical analysis was performed with Prism 10.2.3 (GraphPad Software). For data sets that were not normally distributed, the Mann–Whitney U test was used for comparisons between two groups, and the Kruskal–Wallis test, followed by Dunn’s test, was used for comparisons of multiple groups. The threshold for statistical significance was set at *p* < 0.05. Data are presented as means ± SDs.

## Figures and Tables

**Figure 1 ijms-25-06466-f001:**
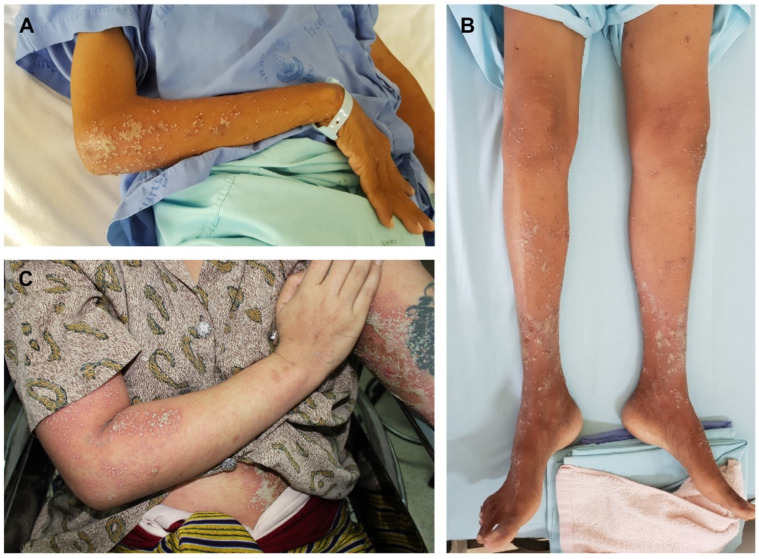
Multiple non-follicular pustules with coalescence on all extremities of an AOID patient (**A**,**B**) and multiple non-follicular pustules with coalescence on all extremities and the abdomen of a patient with generalized pustular psoriasis (**C**).

**Figure 2 ijms-25-06466-f002:**
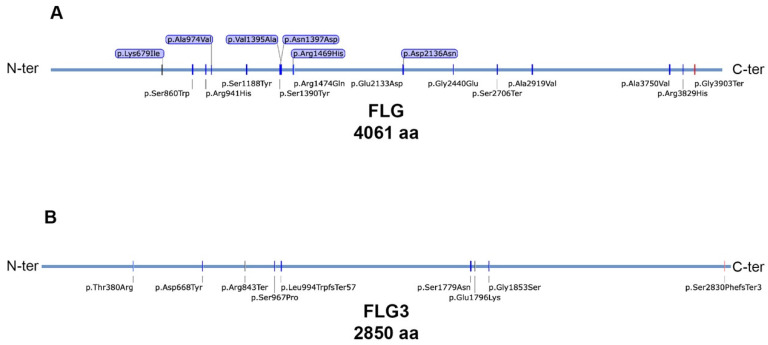
Annotation of the FLG and FLG3 variants identified in this study. The primary structures of FLG (**A**), FLG3 (**B**), and the genetic variants identified in this study were visualized using SnapGene v.7.02. The primary structures of filaggrins were retrieved from Uniprot (FLG: P20930; FLG3: Q86YZ3). Red marks indicate previously unreported variants.

**Figure 3 ijms-25-06466-f003:**
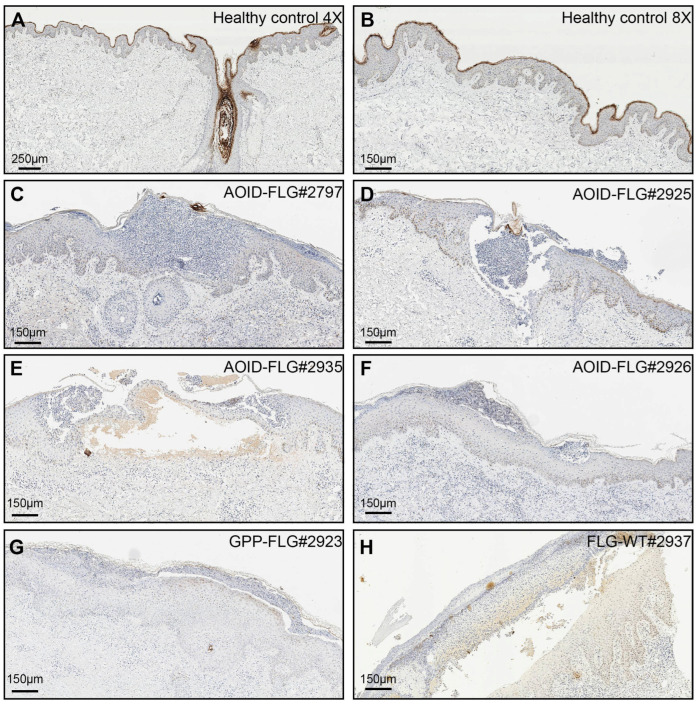
Immunohistochemical comparison of filaggrin expression in healthy control skin to that in skin from anti-interferon-γ autoantibody (AOID) patients with pustular reactions (AOID-PR) and pustular psoriasis (PP) patients who carry FLG variants. Formalin-fixed, paraffin-embedded healthy skin was stained with a filaggrin antibody. Filaggrin was only expressed in well-differentiated keratinized epithelial cells, including hair follicles. (**A**) Healthy control skin (4X). (**B**) Healthy control skin (8X). AOID-PR patients and PP patients displayed reduced filaggrin immunoreactivity compared to healthy controls. (**C**) AOID-PR-FLG-2797 (8X). (**D**) AOID-PR-FLG-2925 (8X). (**E**) AOID-PR-FLG-2935 (8X). (**F**) AOID-PR-FLG-2926 (8X). (**G**) PP-FLG-2923 (8X). (**H**) AOID-PR-FLG-WT (8X).

**Figure 4 ijms-25-06466-f004:**
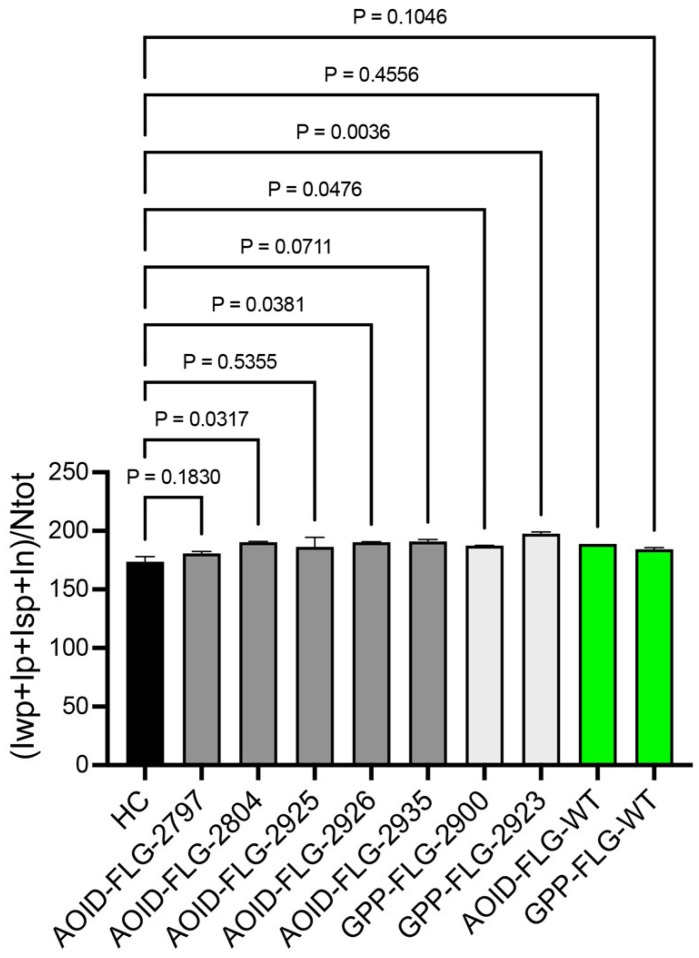
Quantitation of immunoreactive FLG protein in skin biopsies. Bar graph displaying means along with standard deviations (SDs). Mixed-effects analysis was utilized, followed by Dunnett’s multiple comparison test. The immunoreactivity of FLG in AOID-PR and PP was compared to that of healthy controls. Bars without any symbol indicate no statistical significance. HC = healthy control; FLG-WT = wild-type filaggrin. Statistical significance was defined as a *p*-value < 0.05.

**Table 1 ijms-25-06466-t001:** List of filaggrin variants identified.

Case	Condition	FLG	FLG3
2787	AOID-PR	p.Asn1397Asp (rs112252908), p.Val1395Ala (rs527894804), p.Ser1390Tyr (rs139061200)	n.d.
2797	AOID-PR	p.Arg1474Gln (rs200551704)	p.Leu994TrpfsTer57 (rs749162569)
2803	AOID-PR	n.d.	p.Ser1779Asn (rs768542886)
2804	AOID-PR	p.Ser2706Ter (rs542799026)	n.d.
2898	AOID-PR	n.d.	p.Ser967Pro (rs200674313)
2904	AOID-PR	n.d.	**p.Ser2830PhefsTer3**
2911	AOID-PR	p.Arg941His (rs547196696)	n.d.
2914	AOID-PR	n.d.	p.Arg843Ter (rs141263661), p.Asp668Tyr (rs765957331)
2925	AOID-PR	p.Arg3829His (rs145079750)	n.d.
2926	AOID-PR	p.Ser860Trp (rs201661720)	n.d.
2935	AOID-PR	**p.Gly3903Ter**	n.d.
2938	AOID-PR	n.d.	p.Arg843Ter (rs141263661)
2951	AOID-PR	p.Ala2919Val (rs533740963)	n.d.
2877	PP	n.d.	p.Gly1853Ser (rs750289349)
2882	PP	p.Arg1469His (rs145675213)	n.d.
2886	PP	p.Ala3750Val (rs769188915)	n.d.
2895	PP	n.d.	p.Glu1796Lys (rs748246449)
2900	PP	p.Gly2440Glu (rs555272052)	n.d.
2905	PP	n.d.	p.Thr380Arg (rs566122590)
2907	PP	p.Asp2136Asn (rs780793108)	n.d.
2910	PP	p.Ser1188Tyr (rs532746197)	n.d.
2916	PP	p.Lys679Ile (rs369325094)	n.d.
2921	PP	n.d.	p.Arg547Gln (rs769962080)
2922	PP	p.Ala974Val (rs143643121)	p.Gly1259Ser (rs200844006), p.Ser2346Thr (rs753301905)
2923	PP	p.Glu2133Asp (rs142983961)	n.d.
2933	PP	n.d.	p.Arg1834Gly (rs755669550)
2937	PP	n.d.	p.His286Arg (rs368737363)

n.d. = not detected; bold characters highlight variants not reported in the Single Nucleotide Polymorphism database (dbSNP) (https://www.ncbi.nlm.nih.gov/snp/, accessed on 20 March 2022).

## Data Availability

Data are available on request from the authors.

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
