# Peer review of "Rare Filaggrin Variants Are Associated with Pustular Skin Diseases in Asians"

_ijms, 2024, doi:10.3390/ijms25126466_

Round 1

Reviewer 1 Report

Comments and Suggestions for Authors

The Authors present an interesting work of the genetics of reactive pustular forms. However the choice of AOID (instead of AGEP for example) may appear debatable. Moreover, lack of a control group for the genetic part casts doubt on the significance of the reported filaggrin variants.

The reported data may be of value, but the authors need to address or at least discuss these limitations

Reviewer 2 Report

Comments and Suggestions for Authors

Title of the article:

Rare filaggrin variants segregate with pustular skin diseases in Asians

 Manuscript ID:

ijms-3034442

General Comments

Thank you for the opportunity to review your interesting manuscript. I enjoyed reading it; however, in accordance with MDPI Guidelines, I suggest reorganizing the abstract by synthesizing the most relevant information into a single paragraph of about 200 words (see Major Compulsory Revisions). Additionally, I recommend providing a clearer explanation of the statistical methods used by dedicating a separate paragraph to this in your Methods section (see Major Compulsory Revisions). Lastly, I suggest expanding the discussion on certain points (see Major Compulsory Revisions). All things considered, I believe that this manuscript should be suitable for publication only after Major and Extensive Revisions.

- Major Compulsory Revisions

Please, consider reorganizing the abstract into a single paragraph of about 200 words.According to MDPI Guidelines (https://www.mdpi.com/authors/layout)“the abstract contains a summary of the entire paper and can be up to 200 words long with only one paragraph”. 

Please add a full paragraph in the Methods section explaining the statistical analyses performed.

I noticed the presence of Infectious Disease (ID) specialists among the authors. Please consider expanding the discussion to include the possible infective etiology among pustular skin diseases and how this assumption might impact your results.

Author Response

Thank you very much for taking the time to review this manuscript. Please find the detailed responses below and the corresponding revisions/corrections highlighted in the re-submitted files.

Point-by-point response to Comments and Suggestions for Authors

Comments 1: Please, consider reorganizing the abstract into a single paragraph of about 200 words. According to MDPI Guidelines (https://www.mdpi.com/authors/layout)“the abstract contains a summary of the entire paper and can be up to 200 words long with only one paragraph”. 

Response 1: Thank you for pointing this out. However, we have submitted a 200-word abstract as a single paragraph in the current version of this manuscript. Kindly clarify the specific issue that needs to be addressed.

Comments 2: Please add a full paragraph in the Methods section explaining the statistical analyses performed.

Response 2: Thank you for pointing this out. We added a paragraph explaining the statistical analyses ON PAGE 4 line 162-167.

Data Analysis

Statistical analysis was performed with Prism 10.2.3 (GraphPad Software). For data sets that are not normally distributed, the Mann–Whitney U test was used for comparisons between two groups, and Kruskal–Wally’s test followed by Dunn’s test was used for comparisons of multiple groups. The threshold for statistical significance was set at P < 0.05. Data are presented as means ± SD.

Comments 3: I noticed the presence of Infectious Disease (ID) specialists among the authors. Please consider expanding the discussion to include the possible infective etiology among pustular skin diseases and how this assumption might impact your results.

Response 3: We appreciate the reviewer's insightful concern. In this study, we included only reactive/sterile pustular skin diseases for analysis, as described in the first paragraph of the introduction. Infective etiologies were not included in this work. An infectious disease specialist was involved in this study since she primarily cared for AOID patients suffering from opportunistic infections due to their immunocompromised state from neutralizing anti-interferon-γ autoantibodies.

Round 2

Reviewer 1 Report

Comments and Suggestions for Authors

Great job!

Reviewer 2 Report

Comments and Suggestions for Authors

No issue detected.